# Carbides Dissolution in 5Cr15MoV Martensitic Stainless Steel and New Insights into Its Effect on Microstructure and Hardness

**DOI:** 10.3390/ma15248742

**Published:** 2022-12-07

**Authors:** Wenle Liu, Xuelin Wang, Fujian Guo, Chengjia Shang

**Affiliations:** 1Collaborative Innovation Center of Steel Technology, University of Science and Technology Beijing, Beijing 100083, China; 2Yangjiang Branch, Guangdong Laboratory for Materials Science and Technology (Yangjiang Advanced Alloys Laboratory), Yangjiang 529500, China; 3School of Material Science and Engineering, Guangdong Ocean University, Yangjiang 529500, China

**Keywords:** carbides, dissolution, microstructure, variant selection, hardness

## Abstract

The dissolution behavior of carbides in martensitic stainless steel and its effect on microstructure and hardness were investigated by using X-ray diffractometer (XRD) and field emission scanning electron microscopy (FE-SEM) with energy dispersive spectrometer (EDS) and electron backscattering diffraction (EBSD). The results indicated that the microstructure after austenitizing heat treatment and oil quenched consisted of martensite, M_23_C_6_ carbides and retained austenite. The temperature and particle size had great influence on the dissolution of carbides. The EBSD results showed that the twin-related variant pair V1/V2 governed the phase transformation. Meanwhile, the density of high-angle grain boundaries (HAGBs) increased with the increase of austenitizing temperature from 950 to 1150 °C. The hardness test results indicated that the hardness first increased and then decreased with the increase of the austenitizing temperature, and the peak appeared at 1050 °C with a Rockwell hardness value of 59.8 HRC. A model was established to quantitatively explain the contribution of different microstructures to hardness. The contribution to hardness came mainly from martensite. The retained austenite had a negative effect on hardness when the volume fraction was more than 10%. In contrast, carbides contributed less to hardness due to their small content.

## 1. Introduction

Cutting tools play an important role in people’s lives and industrial production [1,2]. High-carbon martensitic stainless steel is used to make knives due to its high strength, high hardness, good wear resistance and good corrosion resistance. This steel, which is designed to be fully austenite at elevated temperatures, is alloyed with between 11.5 and 18.0% chromium and up to 0.6% carbon [3]. 5Cr15MoV martensitic stainless steel is a typical material for making kitchen knives. The as-quenched microstructure of most high-carbon martensitic stainless steels is mainly lath martensite and carbides [4,5].

The carbides are an important phase in high-carbon martensitic stainless steel. The volume fraction, type, size and distribution of carbides affect the microstructure and performance of martensitic stainless strongly [6,7]. Barlow et al. [3] proposed that more dissolution of carbides resulted in higher retained austenite and lager austenite grain size during austenitizing and quenching in 420 martensite stainless steel. Kulkarni Srivatsa et al. [8] indicated that low carbide content and fine carbides along the grain boundary and in the matrix would improve toughness in 13% Cr martensitic stainless steel. Some other works [9,10,11,12] showed that coarse carbides were harmful to mechanical performance, and due to the Cr depletion around carbides, corrosion resistance was poor. Junru Li et al. [9] proposed that large-size M_23_C_6_ carbides, whose size was 0.3–1.5 μm, were notably harmful to the low-temperature toughness of martensitic heat-resistant steels. Bonagani et al. [10] showed that after tempering, the presentation of a Fe-rich surface file and massive carbide precipitation with a Cr depletion zone of 7–9 nm at the carbide interface resulted in the pitting corrosion becoming poor. However, literature [13] indicated that the depleted carbon and chromium content in the matrix also contributed to ductility by increasing the necking deformation needed for void nucleation. A uniform distribution of carbides could improve the toughness of material and reduce the ductile-brittle transition temperature [14]. Yudan Yang et al. [15] found that the carbide dissolution in high-carbon martensitic stainless steel could be distinguished in three stages based on the changes in the content and the average size of the carbides, which displayed a significant impact on performance. However, the study on the dissolution behavior of carbides and the relationship between microstructure and hardness that was quantitatively studied in high-carbon martensitic stainless steel has been seen rarely.

In this study of 5Cr15MoV martensitic stainless steel, carbide dissolution behavior is discussed, and its effect on the microstructure, and the quantitative relationship between microstructure and hardness also, are studied under different austenitizing conditions. The purpose is to present a novel perspective for understanding the role of carbides on microstructure and hardness in high-carbon martensitic stainless steel.

## 2. Materials and Methods

The studied material was commercial 5Cr15MoV martensitic stainless steel after cold rolling and annealing treatment. The dimensions of the cold rolled plates were 1200 mm × 220 mm × 2.5 mm and their chemical composition, which was measured by using a PMI-MASTER PRO direct reading spectrometer, is listed in Table 1. The steel was cut into 15 mm × 15 mm × 2.5 mm by a wire cutting machine. To avoid decarbonization, the vacuum quartz tubes, which were filled with 0.01 MPa argon, were used to protect samples during heat treatment when they were austenitized at 950, 1000, 1050, 1100 and 1150 °C for 10 min. Then, oil quenching was carried out immediately. During oil quenching, the quartz tubes were broken by a hammer in oil immediately.

After quenching, metallographic specimens were ground, polished and etched with 10% sulfuric acid in saturated potassium permanganate solution in order to observe carbides. The microstructure was investigated by TESCAN CLARA GMH field emission scanning electron microscope (FE-SEM) (TESCAN Brno, s.r.o., Brno, Czech Republic) equipped with an Oxford UlitmMax65 energy dispersive spectrometer (EDS) (Oxford instruments, Abingdon, UK) and Oxford Symmetry S2 electron backscattering diffraction (EBSD) (Oxford instruments, Abingdon, UK). Samples for EBSD study were electropolished using a solution containing 87% alcohol, 8% perchloric acid and 5% glycerol. Next, the crystallographic information was analyzed by Channel 5 software (version: 5.12.74.0) from Oxford-HKL (Oxford instruments, Abingdon, UK). MATLAB software (version: 2021b) was used for crystallographic analysis [16,17].

The content of each phase was measured with X-ray diffraction (XRD) using Cu-K α radiation. The scanning angle was 47–93°, and the scanning rate was 2° min^−1^. Afterward, the volume fraction of the retained austenite was calculated by Equation (1) [18].
(1)$$V_{\gamma }=\frac{1-V_{c}}{1+G\frac{I_{\alpha \left(hkl\right)}}{I_{\gamma \left(hkl\right)}}}$$
where $V_{\gamma }$ and $V_{c}$ are the volume fraction of retained austenite and carbides, respectively; G is the ratio of the intensity factors corresponding to austenite and martensite specific crystal faces, as shown in Table 2. $I_{\alpha \left(hkl\right)}$ and $I_{\gamma \left(hkl\right)}$ are the diffraction peak integration intensity of martensite and austenite lattice planes, respectively. $V_{c}$ is taken from experimental data.

The hardness of as-quenched samples was measured by using a Rockwell hardness tester with a load of 150 kg and a loading time of 15 s. Samples for hardness testing were ground to grit size 1000 with silicon carbide paper to guarantee a well-defined surface state.

## 3. Results

### 3.1. Microstructure Characterization of the Annealed Sample

The commercial 5Cr15MoV martensitic stainless steel used in this study was annealed at 800 °C for 36 h. The microstructure characterization result is shown in Figure 1. As shown in Figure 1a,b, it can be clearly seen that the microstructure is composed of a large amount of spherical and ellipsoidal precipitates, ferrite and a small amount of retained austenite. The EDS result shows that these precipitates are Cr-enriched carbides. XRD results shown in Figure 1c indicate that the microstructure consists of ferrite and M_23_C_6_ carbides. The mass fraction of phases in the 5Cr15MoV martensitic stainless steel was calculated by Thermo-Calc using the TCFE7 database. As shown in Figure 1d, below 820.2 °C, the mass fraction of the M23C6 carbides remains unchanged at around 9.3%. Thus, the carbide is identified as (Fe, Cr, Mo)_23_C_6_.

### 3.2. Microstructure Characterization of the Quenched Samples

As can be seen in Figure 2, the microstructure of the as-quenched samples at different austenitizing temperatures is mainly composed of lath martensite, carbides and a small amount of retained austenite. In order to observe the carbides clearly, we used 10% sulfuric acid in saturated potassium permanganate solution to etch samples, in which the carbides are eroded and the martensite matrix is not eroded. However, the morphology of the martensite matrix can be observed clearly in the band contrast (BC) map by using EBSD. In Figure 2, the red arrows point to carbides. As shown in Figure 2, the morphologies of carbides are mainly spherical and ellipsoid, which were inherited from the annealed state [15]. It is well-known that the area fraction equals the volume fraction if the particles distribute uniformly in the matrix [19]. The volume fractions of carbides summarized in Table 3 were counted by using the image processing method based on the SEM images. With an increase of the austenitizing temperature from 950 to 1150 °C, the volume fractions of carbides decrease from 8.82% to 0.25%. Notably, the volume fraction of the carbides decrease rapidly at 1050 °C. At 950 °C, the carbides dissolve slightly, and the prior austenite grain boundaries are indistinct. With an increase of the austenitizing temperature, carbide dissolution accelerates and the size of prior austenite grains also increase. When the austenitizing temperature reaches 1150 °C, the carbides almost disappear. According to the results of EDS shown in Figure 2i and the XRD shown in Figure 3, the carbides are M_23_C_6_ (M = V, Mo, Cr, Fe).

As shown in Figure 2b,d,f,h,j, the martensite lath retained austenite, which is indicated in the red regions that are observed clearly. With an increase in the temperature, the size of the martensite lath becomes finer. Due to the inaccurate volume friction of the retained austenite, by using EBSD results, which were collected in a small area, the XRD measurement was carried out, and the results are shown in Figure 3. At an austenitizing temperature of 950 °C, the diffraction peaks of M_23_C_6_ and martensite are the strongest, while the diffraction peaks of austenite are the weakest. With an increase in the austenitizing temperature, the diffraction peaks of M_23_C_6_ and martensite become weaker, and the diffraction peaks of austenite become stronger. The calculated result of the retained austenite volume fraction by Equation (1) is shown in Table 3. At austenitizing temperature of 950 °C, the volume fraction of retained austenite is 1.95%. At austenitizing temperature of 1150 °C, the volume fraction of retained austenite is 11.10%. In Figure 3, a very pronounced decrease of 2-theta for martensite with the increase of austenitizing temperature can be observed, which corresponds to an increase of lattice parameters, implying more carbon in the martensite. These results indicate that with an increase in the austenitizing temperature, the volume fraction of retained austenite increases because the M_s_ temperature decreases as the number of C and alloying elements increases [3].

### 3.3. Hardness

Figure 4 shows the Rockwell hardness of the samples after oil quenching at different austenitizing temperatures. The results show that the hardness of the experimental steel increases with the increase of the austenitizing temperature in the range of 950–1050 °C, and the hardness increases from 50.3 HRC to 59.8 HRC. However, the hardness decreases from 59.8 HRC to 37.0 HRC as the austenitizing temperature is in the range of 1050–1150 °C, which is due to the softening effect of retained austenite.

## 4. Discussion

### 4.1. Dissolution Behavior of M_23_C_6_

Based on the SEM images, the size and morphology of M_23_C_6_ carbides were investigated at different austenitizing temperatures. The morphology of carbides was mainly spherical and ellipsoid. As shown in Figure 5a, with an increase in the austenitizing temperature, the volume of M_23_C_6_ carbides decreased gradually, and the average diameter of M_23_C_6_ carbides did not decrease monotonically. With an austenitizing temperature in the range of 950–1050 °C, the average diameter remained the same at approximately 0.53 μm. At the austenitizing temperature 1100 °C, the average diameter increased by a small margin to 0.6 μm. When the austenitizing temperature was 1150 °C, the average diameter decreased to 0.28 μm rapidly. As shown in Figure 5b, with an increase in the austenitizing temperature, the frequency of tiny M_23_C_6_ carbides decreased rapidly. This result was basically consistent with the previous study [15]. However, the study [15] indicated that the dissolution of M_23_C_6_ carbides was distinguished in three stages based on the changes in the content and the size of carbides. Firstly, small-sized carbides (<0.5 μm) dissolved dominantly. Secondly, large-sized carbides (<1 μm) dissolved dominantly. Lastly, larger-sized carbides (>1 μm) began to dissolve. These distinctions might not be correct.

According to Figure 1d, at the austenitizing temperature in the range 950–1150 °C, the M_23_C_6_ carbides dissolve. From a thermodynamic point of view, at the beginning of the dissolution of M_23_C_6_ carbides, the thermodynamic conditions for each carbide particle are the same; thus, the dissolution of carbide particles is in no particular order, assuming the composition of the matrix is uniform.

According to study [20], as spherical precipitates became smaller, the dissolution rate increased. For spherical precipitates, the kinetics of dissolution of carbides was assumed on the basis of a diffusion-controlled regime and there was no interaction between carbides. The kinetic formulation of dissolution was developed [20].
(2)$$t_{m}=\frac{R_{0}^{2}}{\left|k\right|D_{v}}$$
(3)$$k=\frac{2\left(C_{I}-C_{M}\right)}{\left(C_{P}-C_{I}\right)}\;$$
where $t_{m}$ is the time of dissolution; $R_{0}$ is the radius of spherical precipitates; $D_{v}$ is the diffusion coefficient of the element; $C_{I}$ is the concentration in the matrix side at the precipitate-matrix interface; $C_{M}$ is the far field composition of the alloy; and $C_{P}$ is the composition of the precipitate.

In 5Cr15MoV martensitic stainless steel, C diffusivity is several orders of magnitude faster than Cr and V, while Cr has the highest fraction among the substituted elements. Therefore, the kinetics of dissolution rate of carbides is assumed to be controlled by the Cr diffusion. According to the literature, the diffusion coefficient of Cr in austenite can be taken from [21]:(4)$$D_{Cr}=10.80\exp\left(\frac{-69700}{RT}\right){\;\mathrm{cm}}^{2}s^{-1}$$

According to Equations (2)–(4), the dissolution behavior of carbides can be discussed in detail. When the initial size of carbides remains unchanged and temperature increases, $D_{Cr}$ and $C_{I}$ increase, which results in $t_{m}$ decreasing. In addition, the temperature is assumed to be constant, and the $t_{m}$ is only determined by the particle size and is proportional to the square of the particle radius. Using the Thermo-Calc with TCFE7 database, the content of Cr in the matrix side at the precipitate-matrix interface was calculated. Thus, the time of carbides dissolution can be calculated. Figure 6 shows the dissolution kinetics of carbides at different temperatures. It is clear that the dissolution time of carbides is proportional to the square of the particle radius, and the higher the temperature, the faster the dissolution.

The above analysis indicates that small particles disappear first and then larger particles dissolve, which is due to the difference in size of the carbide particles. As the dissolution of a large number of fine carbides first results in a decrease of the chromium concentration between the carbides and the matrix, the dissolution rate of larger carbides particles becomes slower.

The M_23_C_6_ carbides act as the sources of carbon and chromium during the dissolution process. At an austenitizing temperature in the range of 950–1150 °C for 10 min, carbon is evenly distributed in the austenite matrix due to the high diffusion coefficient of carbon in austenite. According to the volume fraction of carbides shown in Table 3, the carbon content of austenite at the austenitizing temperature can be calculated by Equations (5)–(7). As oil quenching was carried out immediately after austenitizing, it was considered that the carbon content of martensite was inherited from the austenite. Although there was retained austenite in samples, there was only a small volume fraction after quenching at 950–1100°C. Thus, the carbon content in martensite shown in Table 4 could be roughly considered as the carbon content in austenite.
(5)$$W_{carb}=V_{carb}\times \frac{\rho_{carb}}{\rho_{0}}$$
(6)$$1=W_{carb}+W_{\gamma }$$
(7)$$W_{c}^{0}=W_{c}^{carb}\times W_{carb}+W_{c}^{\gamma }\times W_{\gamma }$$
where $W_{carb}$ and $W_{\gamma }$ are the mass fraction of carbides and austenite, respectively; $V_{carb}$ is the volume fraction of carbides; $\rho_{carb}$ and $\rho_{0}$ are the density of carbides and matrix, respectively; and $\rho_{carb}$ is 6.97 g/cm^3^, $\rho_{0}$ is 7.87 g/cm^3^; $W_{c}^{0}$, $W_{c}^{carb}$ and $W_{c}^{\gamma }$ are the mass fraction of carbon in the matrix, carbides and austenite, respectively.

### 4.2. Crystallographic Analysis of the Microstructure

Figure 7 displays the BC maps and IPF maps of quenched samples. The boundaries in Figure 6 are drawn for misorientation between adjacent points larger than 5°, because the misorientation calculations imply that all of the boundaries between variants should have misorientations larger than 10°. To clearly depict the variant morphology in BC maps, the boundaries are divided into three types according to the misorientation: white lines (5° < θ < 15°), black lines (15 < θ < 45°) and yellow lines (θ > 45°). With an increase in the austenitizing temperature, as well as an increase in the dissolved carbon content, the size of the packet becomes larger, but the lath is refined and the frequency of the high-angle grain boundaries (HAGBs, θ > 45°) increases significantly, as shown in Figure 7k.

The literature [22] studied the effect of carbon content on the variant selection rules, and the grain boundary density showed that in micro-alloy with carbon from 0.03 wt.% to 0.06 wt.%, the twin-related variant pair V1/V2 governed the phase transformation at 0.06C, whereas the misoriented pair V1/V4 was dominated at 0.03C. In order to study the variant selection in 5Cr15MoV stainless steel, the following analysis was carried out. 

It is well known that 24 specific ferrite orientations (variants, V1–V24) could be formed within a single austenite grain in the K-S orientation relationship (OR): {111}γ//{110}α, <110>γ//<111>α [23,24]. However, the actual OR of martensite with respect to an austenite matrix would always deviate some degrees to the exact K-S OR due to the transformation strain. Using iterative numerical calculation methods developed by the authors’ research group based on the Euler angle database from the EBSD results [16,17], the actual average OR of each sample was calculated, which is shown in Table 5. The fraction of 23 variant pairs (V1/V2-V24) was calculated in the samples based on the calculated OR, which is shown in Table 6. To identify the variant selection, the length fraction of inter-variant boundaries between V1 and other variants, we employed the specific calculation method explained in reference [25], which is shown in Figure 8. It can be seen that the V1/V2 variant pairs consistently governed the martensite transformation. With an increase in the temperature, the length fraction of the inter-variant boundary of the V1/V2 variant pair also increased. Study [26] indicated that the increase of carbon led to a decrease of the transformation temperature and an increase of the transformation driving force, such that more V1/V2 variant pairs were obtained to accommodate the transformation strain to improve the hardenability in 0.12C steel and 0.09C steel. Thus, in this study, the fraction of the boundary length of the V1/V2 variant pair that increased with the increase of the temperature was due to the effect of the carbon content from 0.075 wt.% to 0.45 wt.%. This discussion can explain the frequency of high-angle grain boundaries increasing with the increasing temperature. According to references [27,28], there was a near-linear relationship of hardness and density of high-angle grain boundaries. Thus, the hardness would increase with the increasing temperature. However, in this study, the hardness increases first and then decreases with the increase in density of the high-angle grain boundaries. 

### 4.3. Correlation of Microstructure and Hardness

In this study, after oil quenched, the microstructure mainly consisted of martensite, M_23_C_6_ carbides and retained austenite. Different microstructures have different hardness. The hardness of M_23_C_6_ is 1520–1600 HV [29]. The austenite hardness is roughly equivalent to the hardness of the austenitic stainless steel in the annealed condition, whose maximum value usually ranges between 185 and 210 HV [30]. According to the study [31], the hardness of martensite mainly depended on the carbon content in austenite; the alloy element hardly ever affected the maximal hardness of the martensite. In this study, at different austenitizing temperature, the carbon content in the austenite was different due to the dissolution of the M_23_C_6_ carbides, shown in Table 4. The Vickers hardness of martensite has a strong relationship with carbon content. From Figure 4, at 1150 °C, the hardness decreases rapidly due to the high volume of retained austenite, which indicates that retained austenite was a negative factor for hardness. Thus, a simple model was developed to illustrate the contribution of different microstructures to the hardness of the matrix. The model is based on the composite model, which is expressed as Equation (8).
(8)$$HV=HV_{M}\times f_{M}+HV_{carb}\times f_{carb}-k\times HV_{A}\times f_{A}$$
where $HV_{M}$, $HV_{carb}$ and $HV_{A}$ are the hardness of martensite, carbides and austenite, respectively. $f_{M}$, $f_{carb}$ and $f_{A}$ are the volume fraction of martensite, carbides and austenite, respectively. k is a factor that was related to austenite and martensite content. If $f_{A}$ < 10%, k is 0, otherwise k is $f_{M}/f_{A}$. The calculation method of $HV_{M}$ is detailed in literature [31].

As shown in Figure 9, for this study, the experimental results and the calculated results are in good agreement at the range of 1000–1100 °C, although the maximum hardness value differs by 2 HRC. For the results from Ref [32], at 950 °C, the calculated result is larger than experimental result (>5 HRC). This might be sensitivity to the volume of carbides at 950 °C, because the dissolution of carbides is slow at 950 °C. At the range of 1000–1100 °C, the experimental results and the calculated results are in good agreement. As for the results from Ref [33], the experimental results and the calculated results are in good agreement at the range of 950–1100 °C. These suggest that this simple model was still relatively accurate, which showed that the contribution to hardness came mainly from martensite, and a small amount of retained austenite (<10%) had little effect on hardness; however, a large amount of retained austenite (>10%) had a very negative effect on hardness. Due to the small content of carbides, they contributed less to hardness.

## 5. Conclusions

In this study, the behavior of the dissolution of M_23_C_6_ carbides was discussed in detail, and its effect on microstructure was studied. In order to explain the relationship between microstructure and hardness, a sample model was developed. The main conclusions were as follows:After oil quenched, the microstructure of 5Cr15MoV high-carbon martensite stainless steel was mainly composed of lath martensite, M_23_C_6_ carbides and retained austenite. With an increasing austenitizing temperature, the volume fraction of carbides decreased and retained austenite increased. When austenitizing temperature was 1050 °C, the highest hardness was obtained.At the initial stage, the dissolution of M_23_C_6_ carbides particles was in no particular order. The small particles disappeared rapidly first due to their small size. Moreover, the dissolution rate of larger carbides particles would become slow as the dissolution of a large number of fine carbides first resulted in decreasing chromium concentration between carbides and matrix.In 5Cr15MoV high-carbon martensite stainless steel, the martensite phase transformation conformed to K-S rules. The twin-related variant pair V1/V2 governed the phase transformation; meanwhile, the density of HAGBs increased at an austenitizing temperature in the range of 950–1150 °C, i.e., a mass fraction of carbon from 0.075% to 0.45%.The contribution to hardness came mainly from martensite in 5Cr15MoV high-carbon martensitic stainless steel. The retained austenite had a very negative effect on hardness when the volume fraction of retained austenite was more than 10%. In contrast, carbides contributed less to hardness due to their small content.

## Figures and Tables

**Figure 1 materials-15-08742-f001:**
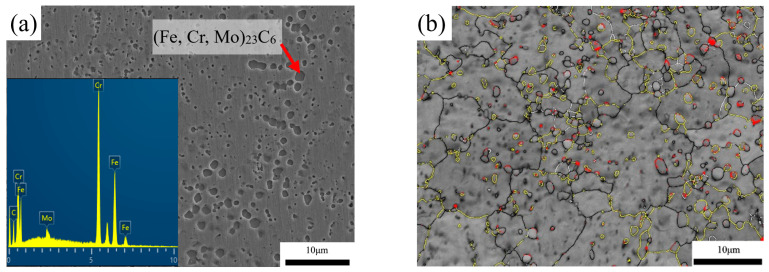
SEM image (**a**), band contrast (BC) map (**b**) and XRD result (**c**) of annealed sample and calculated mass fraction of phases of 5Cr15MoV (**d**).

**Figure 2 materials-15-08742-f002:**
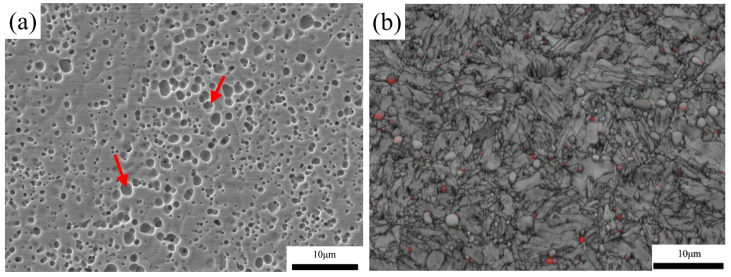
SEM micrographs (**a**,**c**,**e**,**g**,**i**) and band contrast (BC) maps (**b**,**d**,**f**,**h**,**j**) of the microstructure of the high-carbon martensitic stainless steel at different austenitizing temperature: (**a**,**b**) 950 °C; (**c**,**d**) 1000 °C; (**e**,**f**) 1050 °C; (**g**,**h**) 1100 °C; (**i**,**j**) 1150 °C with EDS result of carbides. In the SEM images, the red arrows point to carbides, and in the BC maps, the austenite is expressed by red regions.

**Figure 3 materials-15-08742-f003:**
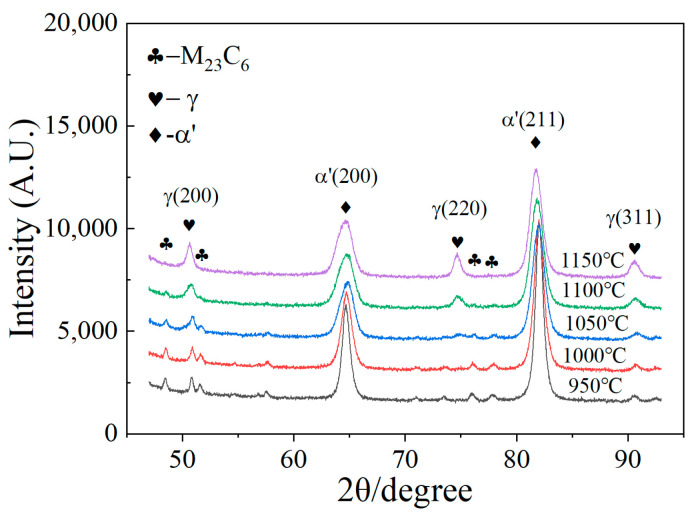
XRD result of the as-quenched samples.

**Figure 4 materials-15-08742-f004:**
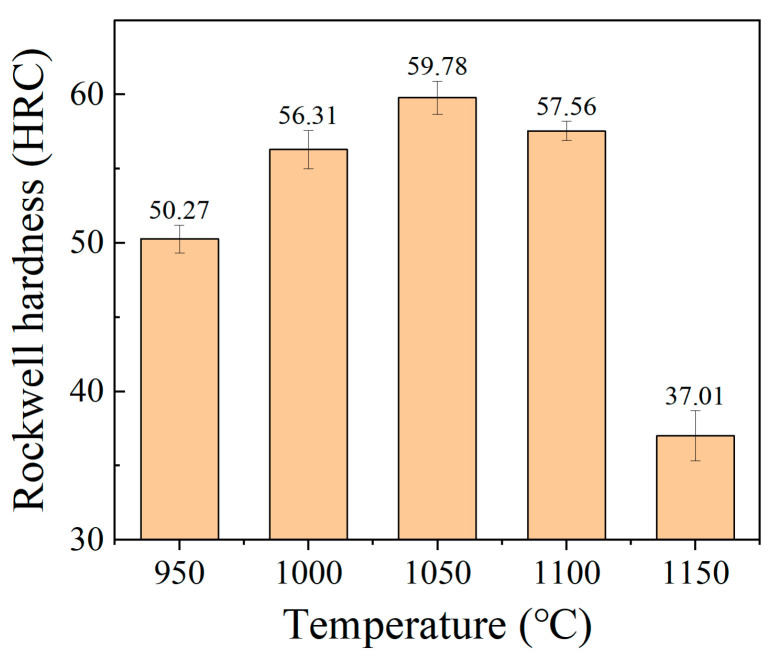
Rockwell hardness after different heat treatment.

**Figure 5 materials-15-08742-f005:**
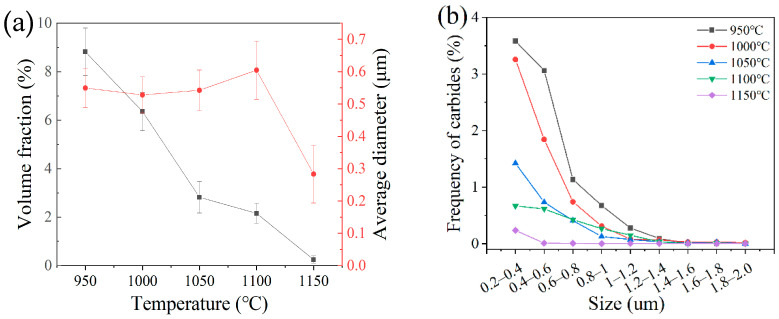
Volume fraction, average diameter (**a**) and distribution frequency (**b**) of M_23_C_6_ carbides at different austenitizing temperatures.

**Figure 6 materials-15-08742-f006:**
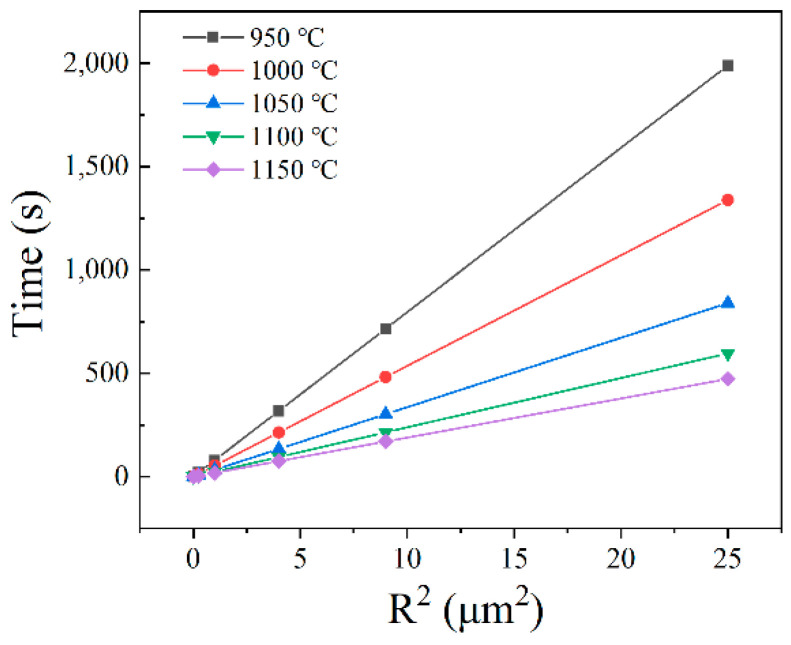
Relationship of dissolution time and radius of carbides.

**Figure 7 materials-15-08742-f007:**
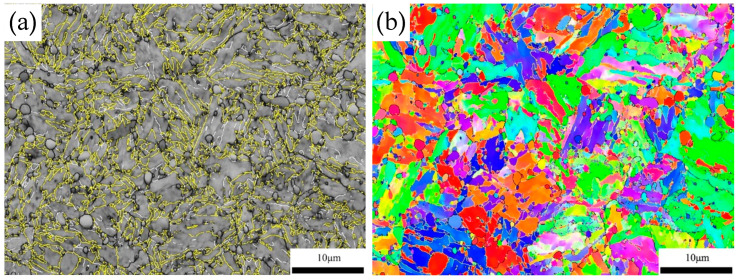
Band contrast (BC) maps with boundary distribution (**a**,**c**,**e**,**g**,**i**) and Inverse pole figure (IPF) maps (**b**,**d**,**f**,**h**,**j**) and frequency of boundaries (**k**). (white line: 5° < θ < 15°, black line: 15° < θ < 45°, yellow line: θ > 45°); (**l**) IPF colouring.

**Figure 8 materials-15-08742-f008:**
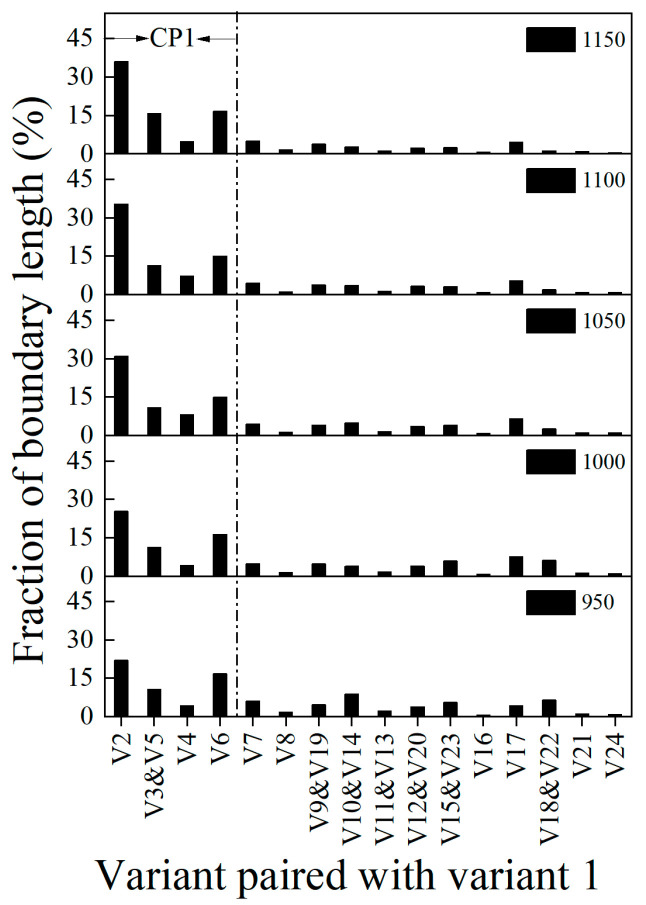
Average length fraction of intervariant boundaries between V1.

**Figure 9 materials-15-08742-f009:**
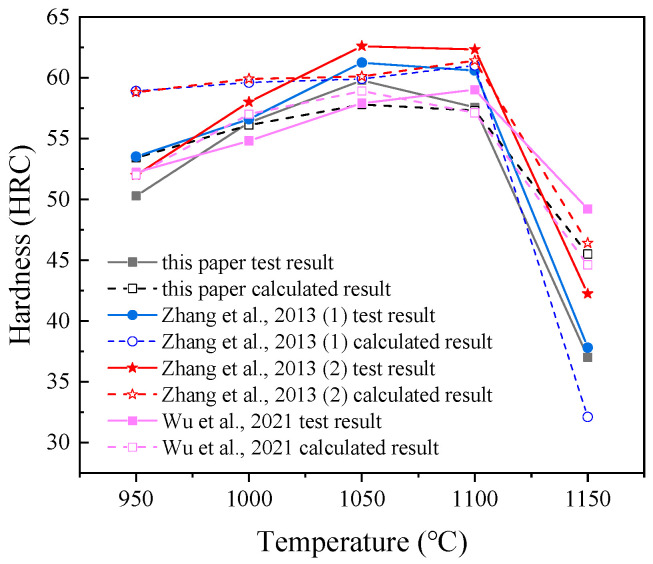
Experimental data of this study and Ref [32,33] and calculation data.

**Table 1 materials-15-08742-t001:** Chemical composition of 5Cr15MoV martensitic stainless steel (wt.%) [15].

Material	C	Si	Mn	P	S	Cr	Mo	V	Fe
5Cr15MoV	0.46	0.41	0.43	0.018	0.012	14.7	0.47	0.21	Bal.

**Table 2 materials-15-08742-t002:** G values used in the calculation of volume fraction of austenite.

Characteristic Diffraction Line	G
(200)γ/(200)α’	2.423
(220)γ/(200)α’	1.274
(311)γ/(200)α’	1.446
(200)γ/(211)α’	1.331
(220)γ/(211)α’	0.6955
(311)γ/(211)α’	0.7939

**Table 3 materials-15-08742-t003:** Volume fraction of carbides and retained austenite at different temperatures.

Phases	Austenitizing Temperature (°C)				
	950	1000	1050	1100	1150
Fraction of carbide (%)	8.82	6.36	2.82	2.16	0.25
Fraction of retained austenite (%)	1.95	2.57	3.28	7.58	11.10

**Table 4 materials-15-08742-t004:** Mass fraction of carbon in martensite.

	950 °C	1000 °C	1050 °C	1100 °C	1150 °C
$W_{c}^{\gamma }\;$(wt.%)	0.075	0.189	0.344	0.372	0.450

**Table 5 materials-15-08742-t005:** Average orientation relationships (OR) of five samples.

		Euler Angle (φ_1_, Φ, φ_2_)
Exact K-S OR		114.2°, 10.5°, 204.2°
Actual OR	950 °C	118.0°, 9.1°, 198.4°
	1000 °C	118.9°, 9.3°, 197.6°
	1050 °C	119.0°, 9.2°, 197.3°
	1100 °C	119.4°, 9.2°, 197.0°
	1150 °C	117.8°, 9.1°, 198.5°

**Table 6 materials-15-08742-t006:** Misorientation axes and angles between V1 and the other variants calculated from the experimentally determined OR (actual OR), and the inter-variant boundary characteristics [22].

Variant	Plane Parallel	Direction Parallel	Rotation Angle/Axis from V1						CP Group	Bain Group	Boundary Type
			Exact K-S OR	950 °C	1000 °C	1050 °C	1100 °C	1150 °C			
V1	(111)_γ_//(011)_α_	[−101]_γ_//[−1−11]_α_	-	-	-	-	-	-	CP1	B1	-
V2		[−101]_γ_//[−11−1]_α_	60.0°/[11−1]	60.2	60.1	60.2	60.2	60.2		B2	Block
V3		[01−1]_γ_//[−1−11]_α_	60.0°/[011]	59.8	60.0	59.8	59.8	59.5		B3	Block
V4		[01−1]_γ_//[−11−1]_α_	10.5°/[0−1−1]	6.3	6.3	5.9	5.8	6.1		B1	Sub-block
V5		[1−10]_γ_//[−1−11]_α_	60.0°/[0−1−1]	59.8	60.0	59.8	59.8	59.5		B2	Block
V6		[1−10]_γ_//[−11−1]_α_	49.5°/[011]	53.9	53.9	54.3	54.3	54.2		B3	Block
V7	(1–11)_γ_//(011)_α_	[10−1]_γ_//[−1−11]_α_	49.5°/[−1−11]	51.8	51.4	51.8	51.7	52.2	CP2	B2	Packet
V8		[10−1]_γ_//[−11−1]_α_	10.5°/[11−1]	8.9	9.3	9.1	9.2	8.6		B1	Packet
V9		[−1−10]_γ_//[−1−11]_α_	50.5°/[−103−13]	52.2	52.1	52.3	52.3	52.2		B3	Packet
V10		[−1−10]_γ_//[−11−1]_α_	50.5°/[−7−55]	51.5	51.2	51.3	51.3	51.5		B2	Packet
V11		[011]_γ_//[−1−11]_α_	14.9°/[1351]	12.9	13.2	13.0	13.0	12.8		B1	Packet
V12		[011]_γ_//[−11−1]_α_	57.2°/[−356]	58.3	58.2	58.0	57.9	58.0		B3	Packet
V13	(−111)_γ_//(011)_α_	[0−11]_γ_//[−1−11]_α_	14.9°/[5−13−1]	12.9	13.2	13.0	13.0	12.8	CP3	B1	Packet
V14		[0−11]_γ_//[−11−1]_α_	50.5°/[−55−7]	51.5	51.2	51.3	51.3	51.5		B3	Packet
V15		[−10−1]_γ_//[−1−11]_α_	57.2°/[−6−25]	57.1	56.8	56.9	56.8	57.2		B2	Packet
V16		[−10−1]_γ_//[−11−1]_α_	20.6°/[11−11−6]	16.5	16.7	16.4	16.4	16.3		B1	Packet
V17		[110]_γ_//[−1−11]_α_	51.7°/[−116−11]	51.3	51.2	51.2	51.1	51.1		B3	Packet
V18		[110]_γ_//[−11−1]_α_	47.1°/[−24−102]	51.2	50.9	51.4	51.4	51.6		B2	Packet
V19	(11–1)_γ_//(011)_α_	[−110]_γ_//[−1−11]_α_	50.5°/[−31310]	52.2	52.1	52.3	52.3	52.2	CP4	B3	Packet
V20		[−110]_γ_//[−11−1]_α_	57.2°/[36−5]	58.3	58.2	58.0	57.9	58.0		B2	Packet
V21		[0−1−1]_γ_//[−1−11]_α_	20.6°/[30−1]	17.6	18.1	17.8	17.9	17.5		B1	Packet
V22		[0−1−1]_γ_//[−11−1]_α_	47.1°/[−102124]	51.2	50.9	51.4	51.4	51.6		B3	Packet
V23		[101]_γ_//[−1−11]_α_	57.2°/[−2−5−6]	57.1	56.8	56.9	56.8	57.2		B2	Packet
V24		[101]_γ_//[−11−1]_α_	21.1°/[9−40]	18.3	18.6	18.3	18.4	18.2		B1	Packet

## Data Availability

The data that support the findings of this study are available from the corresponding author upon reasonable request. Source data are provided with this paper.

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
