# Peer review of "Carbides Dissolution in 5Cr15MoV Martensitic Stainless Steel and New Insights into Its Effect on Microstructure and Hardness"

_materials, 2022, doi:10.3390/ma15248742_

Round 1
Reviewer 1 Report
This manuscript studied the dissolution behavior of carbides in a commercial 5Cr15MoV steel. Microstuctures of samples with different austenization temperatures and quenching process were characterized. This work might be of interest to give insights into microstructure evolution in this type of steels. But, in general, the manuscript is not well written. Much text has to be rewritten and the English has to be thoroughly polished. Several technical points are missing and some discussions are not convincing. Therefore, this work has to be substantially reworked and a major revision is required. Specific comments are listed as follows.
1) Line 42, literatures => works;
2) M23C6-type => M23C6;
3) Line 50, Uniformly => a uniform;
4) Line 59-60, no quantitative relationship, only relationship is quantitatively studied; text should be rewritten.
5) Line 65-71, a commercial; What is the shape or dimensions of the raw material before cutting ? “designed to be” means it is 10 min or the time is different ? What is the oil quenching temperature ? Is it the same for the 5 samples ? Any gas e.g. Ar blown into the quartz tubes ? When quenching, were quartz tubes broken or remain unbroken ? How you control the cooling during quenching ?
6) Equation 1, you should point out the value of the carbide volume fraction will be taken from experimental data; otherwise, it is confusing how you can derive V_gamma.
7) Crystal faces – this is the first time I read this word; shouldn’t it be lattice planes ?
8) Line 95, a loading time of
9) Figure 1d, legend font size and the text is too small; text for scale bars in a and b should be the same.
10) Line 119, well => while or whereas
11) Line 122-124, how you transform the area fraction of carbides to volume fraction ?
12) Line 131, was => is. A lot of tense are mistakenly used throughout this manuscript.
13) Line 144-145, what is the argument for the inaccuracy ? How you index retained austenite with EBSD ?
14) Line 154-155, comes back to my previous question, what is the oil quenching temperature ? And what are Ms temperatures roughly in your samples ?
15) Figure 3, a very pronounced increase of 2-theta for martensite with decreasing austenization temperature can be observed. This corresponds to a decrease of lattice parameters, implying less carbon in the martensite. You should add this point in your discussion.
16) Line 172, improving => increasing
17) Line 174-184, What is the smallest carbides resolved in this study ? This should be clarified.
18) Line 182-184, the text is not accuracte; it should be reformulated to: the smaller carbides dissolve faster than the larger ones.
19) Figure 5b, why is the frequency not normalized ? If the data points are normalized, one would expect no big differences in the size distribution. Only the lower temperature one shift the size a bit towards a larger value.
20) Equation 2 and 3, you should point out the kinetics is assumed on the basis of diffusion controlled regime and there is no interaction between carbides. Ideally, these two equations only apply to one single spherical carbide.
21) Line 197-200, all “was” => “is”; matrix => matrix side
22) Line 201, you should point out that C diffusivity is several orders of magnitude faster than Cr and V, while Cr has the highest fraction among the substitutional elements. Therefore, the kinetics is assumed to be controlled by the Cr diffusion.
23) Line 206, remained => remains; rase => raises or increases
24) Line 207-219, please add a plot to illustrate that the dissolving time is proportional to D^2.
25) Equations 5-7, does it mean no carbon in the martensite ?
25) Line 233-226, the same problem with the tense. Change to the present tense.
26) Line 231, why no misorientations smaller than 5 degrees ?
27) Figure 6k, does it suggest more twins are found with increasing austenization temperature ?
28) Line 250, was => is; Line 257, To identify the variant selection.
29) Table 5, it is confusing. I do not know what those Euler angles correspond to which.
30) For the variant selection, you claim V1/V2 dominates, while in Figure 7, it shows the CP1 takes up the most of the distribution. This is not fully consistent.
31) Line 295-299, past tense should be changed to present tense.
32) Line 307, Line 317, sample model => simple model;
33) Lien 284-285, the hardness of martensite depends on the carbon content in austenite – it is very confusing; do you mean the martensite inheritate the austenite concentration ? However, in this study, as you showed in Equations 5-7, you assumed there is no carbon in the martensite. Please give a clear and explicit explanation.
34) What is the optimal austenization temperature for this specific steel ? It should become one of the conclusions.
Reviewer 2 Report
The manuscript represents good research on the dissolution of carbides in 5Cr15MoV MSS. The research is well-planned and presented, the appropriate equipment was used. The results are presented clearly and with many pictures of microstructure and diagrams, as well as with appropriate explanations of results and detected processes.
According to my opinion, the manuscript should be accepted and a few minor suggestions would be:
- In Section 2, in the beginning, you can add the typical field of application for the used steel,
- Fig. 1d has a very small legend caption, it is almost unreadable,
- In Fig. 4 you presented the hardness of the steel after different heat treatment, however, there is a very big difference in hardness at 1100 and 1150oC. What do you think is the reason? If possible, add one sentence.
Reviewer 3 Report
1. Explain the dissolution behavior of carbides
2. Write the full form of V1/V2.
3. Provide reference of table 1
4. The morphology of carbides was mainly spherical and ellipsoid. Add citation to valid this point.
5. Referer the following paper
https://doi.org/10.1016/j.jnucmat.2020.152376
https://doi.org/10.1016/j.matpr.2019.07.283
Round 2
Reviewer 1 Report
The authors updated the manuscript with a fast but dirty version. Please substantially revise the manuscript again and make a careful revised manuscript.
1) Line 58, is rarely seen
2) line 66, a commercial
3) Line 71, how much argon ? what is the pressure of the argon in the quartz tubes ?
4) What is the oil quenching temperature ? You did not answer this question ! However, in your reply 14, you claim the oil quenching temperature is the same as the austenization temperature. How could that be ? I am totally confused.
5) Line 126 – 128, add how you transform area fraction of carbides to volume fraction. Do not be sloppy at your revisions !
6) Line 135, the carbide is identified as
7) Line 159, an increase
8) Line 189-193, you claim the cited work conclude the three stages for carbide dissolution. However, that is not accurate. All the carbides start to dissolve as long as there is thermodynamic driving force. Whilst small carbides dissolve faster, Large carbides dissolve slower and the changes in size would require more time to be experimentally observed. The text should be completely rewritten.
9) Equations 5-7, if you assume the martensite inherites the composition from the austenite, please write gamma as gamma prime. Even so, these equations are still not correct, because you have retained austenite (~10%). The calculation results in Table 4 can only be considered as very rough estimates.
10) I do not understand your response 27 to my previous question. So I am asking this question again.
11) Your response 29 does not answer my question ! Are those Euler angles correspond to particular parent austenite grains ? or what are they ? Please also clarify this in the text in your new revision.
12) In your response 30, isn’t the CP1 including V1/V2 selection ? Your reponse does not answer my question.
